# Exploring Feedback Mechanics during Experiential Learning in Pharmacy Education: A Scoping Review

**DOI:** 10.3390/pharmacy12030074

**Published:** 2024-05-07

**Authors:** Dania Alkhiyami, Salam Abou Safrah, Ahsan Sethi, Muhammad Abdul Hadi

**Affiliations:** 1QU Health, Qatar University, Doha 2713, Qatar; da097962@student.qu.edu.qa (D.A.); asethi@qu.edu.qa (A.S.); 2Hamad Medical Corporation, Doha 3050, Qatar; ssafrah@hamad.qa; 3College of Pharmacy, QU Health, Qatar University, Doha 2713, Qatar

**Keywords:** feedback, pharmacy education, pharmacy student, pharmacy resident, preceptors, experiential learning

## Abstract

(1) Background: This scoping review aims to explore the literature on feedback for pharmacy students during experiential learning, with a focus on identifying the modes of delivery of feedback and the perceived impact of feedback on student learning outcomes. (2) Methods: The scoping review was conducted in accordance with the Joanna Briggs Institute (JBI) methodology and reported following the Preferred Reporting Items for Systematic Reviews Extension for Scoping Reviews (PRISMA-ScR) guidelines. PubMed, Web of Science, Embase, EBSCO, ERIC, and ProQuest Central were searched electronically from their inception until the end of February 2023 using a combination of keywords and MeSH terms related to feedback, pharmacy education, and student learning outcomes. Data were synthesized narratively. (3) Results: This review included 13 studies published between 2008 and 2022. Almost half of the included studies were conducted in the USA (n = 6, 46%) and reported the perspective of undergraduate pharmacy students (n = 6, 46%). Verbal feedback was the most common mode of feedback delivery (n = 6, 46%). The enablers of effective feedback included timely feedback (n = 6, 46%), feedback provided in a goal-oriented and objective manner (n = 5, 40%), and student-specific feedback (n = 4, 30%). On the other hand, the most common impediments to feedback efficacy were providing extremely positive feedback and lack of constructive criticism. (4) Conclusions: Our findings highlight the importance of feedback model implementation in pharmacy education and preceptor training programs to ensure effective and quality feedback to pharmacy students.

## 1. Introduction

A skilled and competent pharmacy workforce is a crucial aspect of a thriving healthcare system, as it plays a vital role in ensuring the safe and cost-effective use of medication [1]. Experiential learning is a significant component of health professionals’ education [2]. In pharmacy education, experiential learning courses offer real-life experiences within the pharmacy curriculum [3]. Experiential learning courses include specific learning objectives and outcomes with assigned tasks that focus on student application of knowledge and skills in a real-world pharmacy practice setting supervised by a licensed and practicing pharmacist (preceptors) for a defined period [4]. However, in certain situations, other healthcare professionals may serve as mentors, depending on the specific context and learning objectives of the experiential learning program. Effectively providing feedback is considered one of the core functions of a clinical preceptor and a critical step in experiential learning as it facilitates learning and development among pharmacy students [5]. In 2008, Van de Ridder et al. defined feedback as “specific information about a comparison between a trainee’s observed performance and a standard” [6]. Feedback can help students identify their strengths and weaknesses, guide how to improve their skills, and motivate them to engage in the learning process [7]. In recent years, there has been a growing body of literature on feedback in medical education [8]. However, much of this literature has focused on medical learners and specific types of feedback, such as formative assessment or peer feedback. Feedback is a vital component of the learning journey; it helps to minimize the gap between current and target performance levels [9]. Feedback functions as a great tool to support the learner’s development and learning process, thereby contributing significantly to their educational improvements [10]. There is a lack of consensus on the most effective forms of feedback for pharmacy students, especially for experiential learning. There are substantial published studies in the healthcare literature about the evaluation process as a form of assessment, but fairly little is published and known about the feedback process [8]. Evaluation is an essential part of the learning and assessment process; however, it is different from feedback, as outlined by Ende et al., who differentiated between delivering information for improvement (feedback) and involving a judgment of performance (evaluation) [11].

A scoping review by Bing-You et al. conducted on feedback for medical education learners found that feedback is an important component of medical education and that there is a need for more research on how to provide effective feedback to learners [1]. The authors identified several key factors that can affect the effectiveness of feedback, including the timing, frequency, and content of feedback, as well as the relationship between the feedback provider and recipient. They also noted that feedback should be tailored to the individual needs of the learner and that learners should be encouraged to use feedback to reflect on their performance and set goals for improvement. This study focused mainly on medical students and residents (83%).

A preliminary search of the Cochrane Database of Systematic Reviews was conducted, and no current or ongoing systematic reviews of the topic were identified. However, after searching PubMed, several relevant articles were found, including a recent scoping review by Nelson NR (2021), which stated that the literature on pharmacy education feedback lacks depth beyond student perceptions. Furthermore, it stated that the effectiveness and quality of feedback are areas for future research, in addition to post-graduate and interprofessional education, and was limited to articles in English [12]. Moreover, this review explores feedback based on feedback metatheories and an integrative feedback model, including five components: message, implementation, student, context, and agents (MISCA) [13]. The MISCA model facilitates a comprehensive exploration of all aspects of feedback. Each component plays an important role in understanding the dynamics of feedback processes. Considering the interrelated components, the integrative MISCA model offers a comprehensive approach to exploring the feedback process and its impact on student learning and performance.

The objective of this scoping review is to explore the literature on feedback for pharmacy students during experiential learning, with a focus on identifying the modes of feedback delivery, the challenges faced by pharmacy students in incorporating feedback during experiential learning, and the perceived impact of feedback on student learning outcomes.

### Review Question

What are the various modes and models of feedback delivery in experiential learning for pharmacy students and how do they impact the attainment of learning outcomes?

The key objectives are as follows:To explore sources (preceptors, peers, patients, and self-assessment) and modalities (verbal, written, electronic, or simulation-based feedback) of feedback in experiential learning and analyze their effectiveness from the perspective of pharmacy learners.To identify enablers of effective feedback and impediments to feedback efficacy in incorporating feedback during experiential learning.To examine the perceived impact of feedback on attainment of learning outcomes, including knowledge acquisition, skill development, and attitudes, to understand the educational value of feedback in the context of pharmacy education.To map the literature on feedback metatheory, MISCA, across its five components: message, implementation, student, context, and agents.

## 2. Materials and Methods

The proposed scoping review was conducted in accordance with the Joanna Briggs Institute (JBI) methodology and reported following the Preferred Reporting Items for Systematic Reviews Extension for Scoping Reviews (PRISMA-ScR) guidelines [14,15].

### 2.1. Search Strategy

A comprehensive search strategy was developed to identify relevant literature from a variety of databases, including PubMed, Web of Science, Embase, EBSCO, ERIC, and ProQuest Central. A combination of keywords and MeSH terms related to feedback, pharmacy education, and student learning outcomes were used to identify relevant studies (Appendix A). Studies were searched from database inception until the end of February 2023. Studies published in the English language without any study design (qualitative, quantitative, mixed-methods) limitations were included. Review articles, letters, opinion papers, and editorials were also excluded. Two reviewers screened the articles for eligibility using the inclusion criteria detailed below using the PCC model. The PCC model, standing for Population/Participant, Concept, and Context, is a framework commonly used in scoping reviews to guide the formulation of research questions and search strategies [15].

#### 2.1.1. Participants

Studies focusing on the perspectives of pharmacy students and residents regarding feedback received during experiential learning.

Exclusion: Studies primarily focused on feedback in non-experiential learning settings or perspectives from other stakeholders (e.g., educators, preceptors) without direct input from students and residents.

#### 2.1.2. Concept

This review focused on the feedback provided to pharmacy students during experiential learning.

Experiential learning in pharmacy education refers to structured placements integrated into the pharmacy curriculum, offering real-life experiential courses with designated learning outcomes and assigned tasks to enable pharmacy students to practice skills and apply acquired knowledge [4].

Feedback: A formative continuing process of non-judgemental information that guides and helps students or learners to build on their skills, attitudes, and future goals [11].

Exclusion: Studies that focused on other aspects of pharmacy education, such as curriculum design or assessment, as well as studies that focused on feedback from other healthcare professions and limited only to peer feedback were excluded.

#### 2.1.3. Context

Studies were conducted in various settings where pharmacy students and residents undergo experiential learning, including but not limited to community pharmacies, hospitals, clinics, and academic institutions.

Exclusion: Studies involving feedback provided in a non-experiential learning setting were excluded.

### 2.2. Study Selection

All identified citations were collated and uploaded to the EndNote desktop, and duplicates were removed. Titles and abstracts were screened by two independent reviewers for assessment against the inclusion criteria. Potentially relevant and eligible sources were retrieved in full and assessed in detail against the inclusion criteria by two independent reviewers. Reference lists of the included articles were searched for relevant papers to ensure a comprehensive search of the literature. A flowchart of the results was updated throughout the review process to detail the search results, duplicates, and screening results. Reasons for the exclusion of sources that did not meet the inclusion criteria were reported in the scoping review. Any disagreements that arose between the reviewers at each stage of the selection process were resolved through discussion and with an additional reviewer.

### 2.3. Data Extraction

Two reviewers independently extracted data from the included studies using a standardized data extraction tool developed by the reviewers. The extracted data included specific details, including the year of publication, participants, study objective, setting, study design, sample size, feedback model, and perceived student learning outcomes. The draft data extraction tool was piloted, modified, and revised as necessary prior to data extraction.

### 2.4. Data Analysis and Presentation

A descriptive analytical approach was employed to collect, summarize and categorize the literature, including a numerical count of study characteristics (quantitative) and thematic analysis (qualitative). The framework synthesis approach was used to identify key themes and patterns in the data. This review explored the literature on feedback based on feedback metatheories (MISCA) [13]. The MISCA model facilitates a comprehensive exploration of all aspects of feedback. Each component plays a crucial role in understanding the dynamics of feedback processes. By considering these interrelated components, the integrative MISCA model offers a comprehensive approach to exploring the feedback process and its impact on student learning and performance.

A systematic approach was used to extract and analyze the data from these studies. Key information from each study, including the study design, sample size, type and model of feedback, and its impact on student learning outcomes, was extracted using a structured data collection tool. Thematic analysis was also used to identify key themes and patterns in the data, such as the enablers of effective feedback delivery and impediments to feedback efficacy in pharmacy education.

## 3. Results

The results of the search and the study inclusion process are presented in the PRISMA-ScR flow diagram (Figure 1). As shown in the PRISMA-ScR flow diagram, the search retrieved 1343 publications. After removing duplicates (n = 521), titles and abstracts of 822 articles were screened, resulting in 27 full-text articles being retrieved and reviewed for eligibility against inclusion and exclusion criteria. This yielded 13 articles [16,17,18,19,20,21,22,23,24,25,26,27,28] that were subsequently included in this review.

### 3.1. Characteristics of the Included Studies

This review included 13 studies (Figure 1). Studies were published between 2008 and 2022. Almost half of the included studies were conducted in the United States (n = 6, 46%). Community care settings (42%), hospital care settings (31%), and ambulatory care settings (21%) were the most common settings in which studies were conducted. Most studies reported the perspective of undergraduate pharmacy students (46%). The sample size varied considerably among the included studies. Settings ranged from as few as 16 participants to community settings with a sample size of 136 participants.

A summary of the characteristics of the included studies is presented in Table 1. Of the 13 studies, most (62%) used cross-sectional surveys and two used mixed methods. Only one of the 13 studies provided a clear definition of feedback [22], whereas none of the studies discussed the model of feedback in their manuscript. The theoretical framework used was only mentioned in two of the 13 studies [22,25]. The study by Wilbur (2019) employed the cultural dimension models of Hofstede and Hall to understand the feedback encounters and behaviors described by the students. The other study by Jacob (Part 1, 2020) adopted the grounded theory method. Verbal feedback was the most common mode of feedback delivery (n = 6, 46%). Moreover, most studies reported the perceived impact of feedback on student learning outcomes (n = 10, 77%). The perceived impact of feedback reported was improved knowledge, communication skills, and development of clinical and self-management skills.

### 3.2. Enablers of Effective Feedback Delivery, Impediments to Feedback Efficacy, and Proposed Interventions to Improve Feedback Delivery (Table 2)

The majority of the studies reported enablers of effective feedback delivery (n = 12, 92%) and barriers (n = 10, 77%). Enablers of effective feedback delivery included regular and timely feedback (n = 6, 46%), feedback provided in a goal-oriented and objective manner (n = 5, 40%), and student-specific feedback/tailored (n = 4, 30%). Other enablers of effective feedback delivery identified in the review were use of structured rubrics to provide feedback, and the interpersonal characteristics of preceptors including training and interest in providing feedback. The scoping review identified several barriers to receiving feedback. One barrier was providing extremely positive feedback and lacked constructive criticism. Other barriers included lack of feedback, providing short and quick feedback, and lack of recognition of individual performance. Moreover, the proposed intervention to enhance feedback mechanisms involved providing training for preceptors in providing feedback and communication skills and using a structured checklist to assess students’ performance.

### 3.3. Mapping the Reported Findings Using the MISCA Model

The data extracted from the included studies were mapped with the MISCA model to present a comprehensive exploration of all aspects of feedback (Table 3). All included studies provided a clear description of the feedback message (content of feedback), implementation (purpose of feedback), and agent (person who provided the feedback). The agent or the source of feedback was mainly from students (n = 10, 76%), including Advanced Pharmacy Practice Experience (APPE) students, Master (MPharm) students, and residents. In one study, source of feedback was peer feedback. However, some studies did not mention whether the feedback was tailored to the student characteristics and context (timing of feedback). The context (time) of feedback varied among the studies: at the end of the rotation (n = 2, 15%), during the rotation (n = 4, 30%), and after the rotation (n = 3, 23%). The other studies did not provide details on the context of the feedback (n = 4, 30%).

## 4. Discussion

This review has contributed valuable insights into the perceptions of undergraduate and post-graduate pharmacy students regarding feedback during experiential learning with a focus on identifying the modes of delivery of feedback, the challenges faced by pharmacy students during experiential learning, and the perceived impact of feedback on student learning outcomes. Verbal feedback was the most common mode of feedback delivery (n = 6, 46%), yet there was no comparison between different modes of feedback on student learning outcomes. Compared to written feedback, verbal feedback empowers students by prompting self-directed questions, fostering critical thinking, and enabling them to take ownership of their work [30]. Verbal feedback is more feasible in experiential settings because each preceptor has a smaller number of students compared to other teaching settings [12]. Effective feedback strategies include several key components that contribute to supporting students’ learning and growth [31,32,33]. Firstly, understanding the importance of feedback with clear and specific objectives sets the foundation for its value. Timeliness plays a pivotal role, emphasizing the need for feedback to be delivered after a task to maximize its impact. Constructive and actionable feedback is another essential component, as it guides individuals towards improvement by highlighting specific areas for improvement and providing a plan to overcome weaknesses [33]. Emphasizing a commitment to follow-up ensures that the feedback is not a one-time event but an ongoing cycle that supports students’ progress.

The included articles in this review described the enablers of effective feedback delivery and impediments to feedback efficacy. The most frequently reported enabler was providing timely feedback, which was supported by other literature [34] and accreditation standards, for example, the American Society of Health-System Pharmacists Standard 3.4 mandates providing verbal formative feedback [35]. The impediments to feedback efficacy were the lack of constructive criticism and providing extremely positive feedback. In a systematic review focusing on feedback in nursing education, it was emphasized that the feedback should be timely and include both positive and constructive comments [2]. Moreover, the limited time is one of the barriers for the preceptors to provide comprehensive, consistent, and constructive feedback [36]. The proposed intervention to enhance feedback involved providing training for preceptors and the use of a structured checklist to assess students’ performance. An Australian study by Lucas et al. highlighted that standardization of preceptor training was important, particularly in feedback [37].

Despite the rich data obtained, certain gaps persist in the existing literature that warrant further investigation. The feedback provided to pharmacy students is mostly written in summative evaluation, and the quality of feedback is rarely assessed [38]. There is limited understanding of how feedback is used in pharmacy education, particularly during experiential learning [13]. First, the literature lacks a clear, comprehensive, and standardized definition of feedback in pharmacy practice during experiential learning. The definition of feedback in pharmacy education is crucial for establishing consistency, ensuring a standardized understanding and application of this fundamental element across pharmacy educational settings [39]. Feedback in healthcare education differs from general feedback due to the critical nature of the medical environment, complexity of medical knowledge, and professional competence involved. The aim is not to create a new definition but rather to establish a consensus on its application within the healthcare education context. This consensus tailors feedback to specific skills, interdisciplinary collaboration, and patient-centered care. By fostering agreement on these unique aspects, healthcare organizations can promote continuous learning among the learners and professionals and improve patient outcomes [8]. Second, studies exploring the process and impact of feedback in pharmacy education are needed to consider the integrative models such as the MISCA model. A robust feedback model serves as a guiding framework that facilitates effective communication between educators and students [38]. The MISCA model is a comprehensive framework for understanding feedback dynamics and facilitates a thorough investigation of factors influencing feedback effectiveness, including message content, delivery methods, student characteristics, contextual factors, and agent attributes. By identifying enablers of effective feedback delivery and impediments to feedback efficacy, the feedback delivery and utilization can be optimized to advance educational practice and enhance student learning outcomes. Moreover, the MISCA model can serve as a framework for future research and practical ideas, guiding the development of evidence-based strategies to improve feedback practices in educational settings [13]. A well-structured feedback model provides a systematic approach for preceptors to communicate constructive insights, enabling students to understand their strengths and identify areas for improvement with a clear action plan. Models of feedback are crucial in the field of medical and pharmacy education as they provide structured frameworks that enhance precision, ensuring effective communication and improvement strategies for students. Moreover, this model contributes to the overall quality of pharmacy education by promoting consistency and transparency in the feedback process. There is also a need for the use of a theoretical framework in pharmacy education research to assist researchers in better understanding the feedback process and guiding the implementation of the proposed intervention. This can result in an effective and sustained feedback process during experiential learning [28]. Implementation of a theoretical framework and feedback models helps in measuring the impact of feedback on student outcomes and understanding the strategies that can enhance student performance and learning [28]. The lack of consistent, constructive, timely, and individualized feedback, as reported by most of the included studies in this review, highlights the need for feedback model implementation in pharmacy education.

### 4.1. Strengths and Limitations

To the best of our knowledge, this scoping review is the first to map studies using the MISCA model to present a comprehensive exploration of all aspects of feedback in the literature. It is important to highlight that most studies were conducted in the USA and Europe, in addition to including studies from Africa and one study from the Middle East, as experiential learning experiences are different in each country. One limitation is that the review was restricted to primary research articles published only in English, which may have limited our findings.

### 4.2. Further Research and Recommendations

Future research in pharmacy education should focus on investigating the implementation and impact of effective feedback models in experiential learning. In addition, research focuses on the successful implementation of the proposed interventions to improve feedback. Equally important is exploring preceptors’ training programs to ensure that they are well-equipped to provide constructive feedback and action plans for improvement and follow-up during experiential learning. These areas require continuous exploration to enhance the feedback process and improve the quality of pharmacy education that promotes students’ development and satisfaction.

## 5. Conclusions

This review provided a wide breadth of literature on the perceptions provided by pharmacy students regarding their experiences of receiving feedback during experiential learning that was mapped with the MISCA model. Our findings highlight the importance of feedback model implementation in pharmacy education and preceptor training programs to ensure effective and quality feedback to pharmacy students. Future research should focus on investigating the implementation and impact of effective feedback models in experiential learning.

## Figures and Tables

**Figure 1 pharmacy-12-00074-f001:**
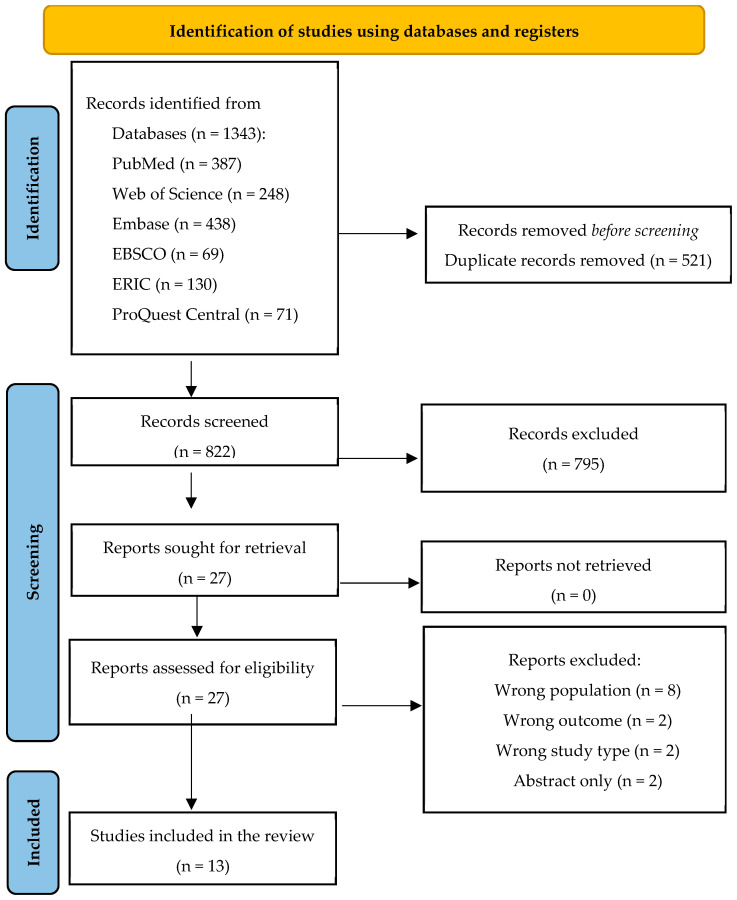
PRISMA flow diagram indicating the study selection process. From Page, “The PRISMA 2020 statement is an updated guideline for reporting systematic reviews” [29].

**Table 1 pharmacy-12-00074-t001:** Summary of the characteristics of the included studies (n = 13).

Author, Year of Publication	Participants and Setting	Study Objective	Study Design and Sample Size	Modes of Feedback	Student Learning Outcomes	Feedback Model and Theoretical Framework
Hyvarinen, 2008 [16]	Undergraduate students in a community setting	Analyze Finnish students’ opinions of the feedback given in patient counseling training.	Qualitative study involving 136 students	Verbal as a discussion	Developing communication skills	Not described
Boland, 2014 [17]	Undergraduate and postgraduate students in community and teaching hospitals	Implement a new process for using student evaluations in developing and evaluating pharmacy residents as preceptors.	Prospective study by 23 pharmacy students for 8 residents.	Written as evaluations	Fostering preceptor development in the preceptor roles	Not described
Bates, 2016 [18]	Undergraduate and postgraduate students in acute care oncology practice	Explore the use of pharmacy learners as a means to expand pharmacy services in a layered learning practice model (LLPM).	Longitudinal study of 16 learners	Verbal through micro-discussion	Improved clinical time management skills, and development of clinical and self-management skills	Not described
Belachew, 2016 [19]	Undergraduate students in community pharmacy in Ethiopia	Investigate the overall experiences of clinical pharmacy students during their clinical attachments and to understand the breadth and depth of clinical skills provided by their preceptors.	A cross-sectional study by 58 students.	Not described	Not described	Not described
Melaku, 2016 [20]	Undergraduate students in the community in Ethiopia	Compare the perceptions of pharmacy clerkship students and clinical preceptors regarding preceptors’ teaching behaviors and feedback provision.	Cross-sectional study by 126 students	Verbal as a discussion	Not described	Not described
Linedecker, 2017 [21]	Undergraduate students in ambulatory care and preceptors.	Evaluate the usefulness of the direct observation of procedural skills rubric in evaluating student performance and clinical skills during ambulatory care rotations.	Cross-sectional by 47 students.	Written and verbal	Enhancing the clinical and communication skills	Not described
Wilbur, 2019 [22]	Undergraduate students enrolled in a Canadian-accredited cross-border pharmacy program in Qatar.	Determine non-Western situated health professional student experiences and preferences for feedback in workplace-based settings.	Focus groups of 27 students	Verbal and written	Cultural influences on student feedback experiences: collectivism, power distance, and context	Cultural dimension models by Hofstede and Hall were employed.
Schweiss, 2019 [23]	Postgraduate students in ambulatory care settings	Implement and evaluate a pharmacy resident documentation peer review process.	Peer review process model that included 25 residents	Written feedback	Improved patient care documentation, providing peer feedback, and the importance of effective interprofessional communication in clinical decision making	Not described
Jacob, 2020 [24]	Undergraduate students in community and hospital settings	Obtain students’ perceptions and feedback on the experiential learning (EL) programs.	Cross-sectional survey with 121 responses	Not described	Not described	Not described
Jacob Part 1, 2020 [25]	Postgraduate students in the community and hospital settings	Obtain feedback from graduates on EL placements and assess the effectiveness of EL in preparing them for pharmacy practice.	Cross-sectional survey with 63 responses, ten one-on-one semi-structured interviews, and a focus group discussion	Not described	Not described	A grounded theory method was adopted.
Jacob Part 2, 2020 [26]	Postgraduate students in the community and hospital settings	Obtain feedback from graduates on their EL and assess the effectiveness of EL in preparing them for pharmacy practice.	Cross-sectional survey with 63 responses, ten one-on-one semi-structured interviews, and a focus group discussion	Not described	Not described	Not described
Hatcher, 2022 [27]	Postgraduate students in ambulatory care and community pharmacy	Describe the development and implementation of a remote required ambulatory care and required community pharmacy dual-cohort Advanced Pharmacy Practice Experience (APPE) rotation from the student pharmacist perspective.	Cross-sectional study using electronic survey, 24 completed the survey	Verbal and peer feedback	Improved abilities on key Center for the Advancement of Pharmacy Education (CAPE) outcomes	Not described
Margolis, 2022 [28]	Undergraduate students in acute and ambulatory care	To determine the appropriateness and feasibility of implementing the Individual Teamwork Observation and Feedback Tool (iTOFT) in APPEs to allow direct observation and rating of students’ interprofessional teamwork skills.	Cross-sectional using a survey of 149 evaluations	Written	Enhanced preceptor feedback for students on interprofessional collaboration	Not described

**Table 2 pharmacy-12-00074-t002:** Enablers of effective feedback delivery, impediments to feedback efficacy, and proposed interventions (n = 13).

Author, Year of Publication	Enablers of Effective Feedback Delivery	Impediments to Feedback Efficacy	Proposed Interventions
**Hyvarinen, 2008** [16]	Committed and trained mentors Mentor’s interest Introduction to the study plan and guidelines Delegation of training tasks	Short feedback discussions Lack of critical and constructive feedback Feedback highlights mistakes only Not familiar with guidelines Providing only positive feedback	Train students to explain the use of the guidelines to their mentors. Mentors need training in analyzing communication skills and providing constructive feedback.
**Boland, 2014** [17]	Good learning experience encouraged residents to take initiative in learning opportunities.	Not receiving formal feedback on their precepting skills Lack of training in teaching abilities and precepting skills Lack of confidence due to limited practice experience	Using student evaluations to develop precepting skills Individual surveys are built for each resident, allowing for personalized feedback Regular meetings with the primary preceptor develop a strategy to improve their precepting skills
**Bates, 2016** [18]	Feedback is provided in a goal-oriented and objective manner. Feedback delivered sensitively ensures that learners feel supported. Learner and preceptor working together to create common goals Regular reflective sessions Providing constructive feedback that focuses on specific areas of improvement Using rubrics ensures consistent and objective evaluations. Offering feedback in real-time as practice experience activities occur	Lack of structured feedback Limited time was dedicated to reflecting on the experience. Feedback was not comprehensive. The feedback had a limited diversity of perspectives.	Use of a structured practice experience continual feedback throughout the experience Provide feedback in a process called feed-forward in a goal-oriented, objective, performance-based, and sensitive style Scheduled reflective sessions, followed by a formal end-of-experience evaluation
**Belachew, 2016** [19]	Timely feedback	Not described	Emphasis should be placed on preceptor training as a crucial component in providing feedback.
**Melaku, 2016** [20]	Preceptors provided practical responsibilities to students. Preceptors explained the goals and expectations to the students. Preceptors are perceived to demonstrate sensitivity and supportiveness towards students. Preceptors closely supervised students. Preceptors provided students with the opportunity to ask, discuss, and exchange opinions. Preceptors spent sufficient time with students. Preceptors were accessible. Preceptors discussed the practical application of knowledge and skills with students.	Subjectivity of feedback Lack of confidence in the evaluation system and preceptors’ ability to provide feedback Students’ dissatisfaction with the instructors’ ability to motivate them	Short-term training is warranted for preceptors. Preceptors should participate in workshops involving the development and implementation of new guidelines.
**Linedecker, 2017** [21]	The DOPS rubric was found to be a practical tool. The use of the DOPS method allowed for both formative and summative assessment of student learning.	Inconsistencies in the feedback provided	The use of a structured checklist to assess students’ performance in areas such as communication, physical examination, and professionalism
**Wilbur, 2019** [22]	Preceptors spent sufficient time with students and provided more credible and valuable feedback. Students preferred receiving feedback in a timely manner. Students appreciated receiving negative feedback along with suggestions for improvement.	Lack of recognition and acknowledgment of students’ performance Preceptors were unwilling to accept feedback for improvement. Lack of documentation of feedback on the written evaluation report Lack of privacy	Development of “near-peer” teaching programs The need for purposeful evaluation of educational interventions in workplace-based settings
**Schweiss, 2019** [23]	Written feedback was more beneficial than Likert-type scale ratings. Allow residents to self-select the notes they want to be reviewed and receive feedback on	Lack of clinical input in the feedback process. Extremely positive and lacking constructive criticism Involving many reviewers is tedious and challenging to manage.	Documentation should include detailed and clear plans for patient care.
**Jacob, 2020** [24]	Quality assurance measures are important to ensure that tutors are qualified and capable of providing effective feedback. Providing monetary compensation to tutors for their time and effort	Community placements did not provide them with enough time to complete their own learning objective. Lack of teaching and learning opportunities Students expressed dissatisfaction with the limited duration of the rotation. Workload was not carefully planned and balanced.	Tutor training programs such as the Preparation for Facilitating Experiential Learning Training (PFEL), can help tutors develop the necessary skills to provide feedback. Universities should signpost relevant support staff that tutors can access if they face challenges or have questions while tutoring students. Have quality assurance measures in place to provide students with an effective and equitable placement experience.
**Jacob Part 1, 2020** [25]	Not described	Not described	Implementation of mandatory training Tutors and placement sites should recognize the value of having students. Further evaluations should be undertaken to determine the amount of placement time required to make students practice-ready.
**Jacob Part 2, 2020** [26]	Feedback should be tailored according to each student’s needs. Feedback should be dialogical rather than transmission-centered. Feedback should be formalized. Continuous quality improvement processes are important to ensure that all students have a standard experience across different sites. This involves analyzing feedback from students, implementing changes if necessary, and closing the loop by providing feedback about the changes or actions implemented.	Lack of feedback	Need for quality assessment of placement sites and tutors
**Hatcher, 2022** [27]	Longitudinal feedback with opportunities to demonstrate improvement	Lack of consistency in feedback provided by different faculty members Remote delivery of feedback may not be as effective; it may limit the interaction between students and faculty. Remote delivery of feedback may result in delays in providing timely feedback to students.	Emphasis on high-touch, high-engagement activities that promote active discussion and consistent feedback for learning Using interactive tools such as MyDispense^®^ and Anticoag Games, and case presentations facilitated by various preceptors Offer orientation sessions for faculty, preceptors, and facilitators involved in the rotation.
**Margolis, 2022** [28]	The iTOFT activity provided a formal structure for feedback on interprofessional teamwork.	Not described	Use validated interprofessional assessment tools

**Table 3 pharmacy-12-00074-t003:** Mapping the reported findings with the MISCA model (n = 13).

Author, Year of Publication	Message (Content)	Implementation (Purpose)	Student Characteristics	Context (Time)	Agent (Self, Peers, Preceptors)
Hyvarinen, 2008 [16]	Guidelines for giving feedback on communication skills and patient counselling training	To help pharmacy students develop their communication skills systematically in real customer service situations.	Not described	At the end of the 3-month training period	Mentors
Boland, 2014 [17]	Presenting skills, abilities in instructing, modelling, coaching, and facilitating	To evaluate and provide valuable information on the precenting skills of pharmacy residents, ultimately fostering their growth and development in this role.	Not described	Feedback provided within one week of the rotation	Co-Preceptors (residents)
Bates, 2016 [18]	Focused on micro discussion, standardized feedback (e.g., rubrics), and cooperative learning to enhance educational gain through clinical activities including medication histories and patient counselling sessions	To explore the use of pharmacy learners to expand pharmacy services in a layered learning practice model (LLPM).	The preceptor tailored the feedback based on the student’s characteristics through reflection meeting.	Feedback was given in real time as practice experience activities occurred.	Preceptors and residents
Belachew, 2016 [19]	Students’ experiences, satisfaction, and perceptions regarding their training program and the performance of their preceptors	To assess the quality of practical skills received by clinical pharmacy students during their clerkship training and to evaluate the abilities of their primary preceptors in providing clinical skills during the clerkship.	Not described	Feedback was obtained at the end of final-year pharmacy students who had undergone clerkship training.	Preceptors
Melaku, 2016 [20]	Students’ strengths and limitations in clinical practice set criteria for student performance.	Increase students’ efficiency and provide students with guidance and support in improving their clinical skills and knowledge.	Not described	Not described	Preceptors
Linedecker, 2017 [21]	Evaluations of the student’s communication skills, patient work-up, critical thinking abilities, patient interviews, and patient education The feedback also highlights areas that require improvement and provides examples of good skills.	To assess the students’ performance, evaluate their readiness for advanced pharmacy practice experiences, and determine if they meet the expectations of a P-4 student.	The feedback is tailored to the students’ level of knowledge and skill, assessing their ability to perform tasks independently.	Feedback was provided after the completion of the (DOPS) exercise.	Preceptors
Wilbur, 2019 [22]	Three themes are associated with cultural influences on student feedback experiences: (1) collectivism, (2) power distance, and (3) context.	To guide students’ ongoing development, understand their performance, gain elaboration on their rated performance, and improve their skills.	Not described	Not described	Preceptors
Schweiss, 2019 [23]	Comments, written feedback, and Likert-type scale ratings on each section of their documentation.	The feedback provided guidance on how to improve patient care documentation, provide peer feedback, and emphasize the importance of effective interprofessional communication in clinical decision making.	Not described	Feedback was given during the residency program quarterly, then reduced to semi-annual reviews to allow for a more thorough and thoughtful review.	Preceptors and peers
Jacob, 2020 [24]	The survey assessed students’ perceptions of various aspects of the EL program including the effectiveness of the EL, tutors and placement sites, and the organization and structure of the EL.	To ensure that tutors are aware of the responsibilities and expectations.	Not described	Not described	Tutor
Jacob Part 1, 2020 [25]	The survey contained questions assessing graduates’ perceptions of the effectiveness of the EL, its organization and structure, as well as feedback on tutors and placement sites.	The feedback identified gaps in the structure and design of the EL component and to gather insights on how to improve the EL experience.	Not described	Feedback was collected after the graduates had completed their MPharm program and were undergoing their pre-registration training.	Tutor
Jacob Part 2, 2020 [26]	Their comments on the experiences observed provide an opportunity for tutors to identify and correct things the students may have misunderstood.	To provide students with an experiential base for reflection.	Not described	Not described	Tutor
Hatcher, 2022 [27]	Application of knowledge through activities such as topic discussions and patient case presentations	To assess the impact of the remote required ambulatory care and required community pharmacy dual-cohort (APPE) rotation on students’ ability to meet the (CAPE) Outcomes.	Not described	Feedback was collected at the end of the rotation.	Preceptors
Margolis, 2022 [28]	The feedback provided to students using the Individual Teamwork Observation and Feedback Tool (iTOFT) focused on their interprofessional collaboration skills.	The feedback provided to students using iTOFT focused on their interprofessional collaboration skills.	PharmD students completing advanced pharmacy practice experiences (APPEs)	Feedback was provided during required acute care and ambulatory care APPEs.	The feedback was given by preceptors who directly observed students’ behavior on interprofessional teams.

## Data Availability

The data presented in this study are presented in the tables.

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
