# Peer review of "Exploring Feedback Mechanics during Experiential Learning in Pharmacy Education: A Scoping Review"

_pharmacy, 2024, doi:10.3390/pharmacy12030074_

Round 1

Reviewer 1 Report

Comments and Suggestions for Authors

Thank you for the opportunity to review this interesting scoping review. I hope my feedback is helpful.

ABSTRACT:

-line 12: includes a focus on “the challenges faced by pharmacy students during experiential learning”. I don’t think this is quite what the paper has set out to do and suggest this statement needs modifying to ensure the focus is on feedback rather than the entire experiential learning process (also in line 76; explained better in lines 90-91)

-line 20: I query use of the term “Most…”6/13 studies is not most. More papers were conducted in the USA than any other country might be a better way to express this? (also in line 242)

-line 22: it is not really clear what facilitators and barriers refer to—is it effectiveness of receiving feedback? I found this lack of focus throughout the paper made it difficult for the reader to determine what the reported outcomes were

INTRODUCTION:

-line 36: I question whether a preceptor must always be a pharmacist?

-line 51: what is meant by the term “evaluation process”. Is Ende referring to assessment?

-lines 55-57: I’m unsure what is meant by this sentence, and whether “therefore:” is an appropriate bridging word—how does this sentence link to the previous one?

-line 79: the review question seems not entirely aligned with the stated objectives in lines 74-77 and is different again from the research question stated in lines 430-431

-line 82: I am not convinced this review focuses on how feedback is “used to enhance the quality of pharmacy education”?

-line 92 rather than impact on learning outcomes, should this be impact on attainment of learning outcomes?

-line 95: a case is not made for why MISCA was chosen. Also it should be spelled out and referenced here. This is better explained in 177-184 however no references are provided in terms of its validated use in other feedback studies which would be helpful to the reader

METHODS

-line 116: what is the PCC model?

-I am concerned the search strategy detailed in appendix A did not detect more literature on Workplace Based Assessments (WBA) and Competency Based Education (CBE) which are becoming more prominent in pharmacy education, and have a significant focus on feedback in experiential settings

-line 120: was the focus solely on “perspectives of pharmacy students and residents regarding feedback …” Did you also look at the nature and models of feedback, etc (relating to the key objectives lines 86-96)?

-line 141 excludes peer feedback but this is specifically included in line 86? This also seems to be the focus of (for example) reference 25

RESULTS

-line 219: why were 795 records excluded?

-lines 249-259: referencing is absent/inconsistent

-lines 253 and 255 refer to “this study” and “the other study” but the actual studies being referred to are not clear to the reader

-line 258 refers to “perceived impact” which I think equates in the table to “student learning outcomes” but this is unclear

-lines 263-274: the wording in this paragraph is very unclear to the reader. Are these barriers and facilitators to receiving feedback, incorporating feedback, preferences, etc for example? Do they relate to giving and/or receiving feedback? How is “lack of feedback” a barrier to receiving feedback?

-table 1 appears to include some studies about feedback from learners relating to their experiential placement rather than a focus on receiving feedback from a preceptor on placement. How does data such as this contribute to an understanding of effective models of feedback?

-table 3: my understanding is the agent (person providing the feedback) in this instance should be the preceptor. In a number of cases the agent is the learner themselves which I think means they provided feedback on the program, rather than feedback to the learner which is the intended focus of this review?

DISCUSSION

-line 349: the prior studies referred to may need to be referenced?

-line 355: I do not believe a strong case has been made to support the need for a specific definition of feedback in the pharmacy experiential learning setting. Why would this need to be different to a general definition of feedback, and those used in education of other healthcare professionals?

-line 359: I do not believe a strong case has been made to support the assertion that studies need to incorporate the MISCA model

-line 367: what does “this model” refer to? MISCA?

-line 369: I do not disagree with this statement but it is somewhat out of place in this paper. One example is given, a better inclusion might be a discussion of what models are available and suitable to use in this situation, with references in support?

Comments on the Quality of English Language

-

Author Response

Reviewer’s comments

Comments and Suggestions for Authors

Thank you for the opportunity to review this interesting scoping review. I hope my feedback is helpful.

ABSTRACT:

Reviewer 1: -line 12: includes a focus on “the challenges faced by pharmacy students during experiential learning”. I don’t think this is quite what the paper has set out to do and suggest this statement needs modifying to ensure the focus is on feedback rather than the entire experiential learning process (also in line 76; explained better in lines 90-91)

Authors’ response: Thank you for your feedback and for highlighting the need for clarity regarding the focus of our paper, the sentence was modified accordingly.

“the challenges faced by pharmacy students in incorporating feedback during experiential learning, and the perceived impact of feedback on student learning outcomes.” (lines 12, 84)

Reviewer 1: -line 20: I query use of the term “Most…”6/13 studies is not most. More papers were conducted in the USA than any other country might be a better way to express this? (also in line 242)

Authors’ response: Thank you for your comment, the sentence was modified as suggested.

“Almost half of the included studies were conducted in the USA (n=6, 46%)” (lines 20,272)

Reviewer 1: -line 22: it is not really clear what facilitators and barriers refer to—is it effectiveness of receiving feedback? I found this lack of focus throughout the paper made it difficult for the reader to determine what the reported outcomes were

Authors’ response: Thank you very much for your comment. We have now considered this comment very carefully and we agree that these actually do not describe facilitators and barriers. Therefore, we have now used the terms enablers of effective feedback and impediments to feedback efficacy as standard terms throughout the manuscript.

INTRODUCTION:

Reviewer 1:-line 36: I question whether a preceptor must always be a pharmacist?

Authors’ response: The role of a preceptor in pharmacy education traditionally aligns with the expertise and knowledge of a pharmacist. However, in certain situations, individuals from other healthcare professions may serve as mentors, depending on the specific context and learning objectives of the experiential learning program. For this review and to reflect common practice, the included studies focus on preceptors who are pharmacists.

“However, in certain situations, other healthcare professionals may serve as mentors, depending on the specific context and learning objectives of the experiential learning program.” (lines 37-39)

Reviewer 1: -line 51: what is meant by the term “evaluation process”. Is Ende referring to assessment?

Authors’ response: The term "evaluation process" refers to the form of assessment. Ende's distinction between feedback and evaluation highlights that while both processes involve providing information, evaluation typically includes a judgment of performance.

The sentences were revised and adjusted according to your suggestion to make clearer to readers.

“There are substantial published studies in the healthcare literature about the evaluation process as a form of assessment, but fairly little is published and known about the feedback process [8]. Evaluation is an essential part of the learning and assessment process, it's different from feedback, as outlined by Ende et al., who differentiate between delivering information for improvement (feedback) and involving judgment of performance (evaluation)” (lines 55-59)

Reviewer 1:-lines 55-57: I’m unsure what is meant by this sentence, and whether “therefore:” is an appropriate bridging word—how does this sentence link to the previous one?

Authors’ response: We agree with you, the sentence was removed as it is not related to the previous sentences. We have rewritten that part of paragraph to make it more clearer.

Reviewer 1: -line 79: the review question seems not entirely aligned with the stated objectives in lines 74-77 and is different again from the research question stated in lines 430-431

Authors’ response: We have now reworded both research question and research objectives to better align them

Reviewer 1: -line 82: I am not convinced this review focuses on how feedback is “used to enhance the quality of pharmacy education”?

Authors’ response: We have now reworded both research question and research objectives to better align them

Reviewer 1: -line 92 rather than impact on learning outcomes, should this be impact on attainment of learning outcomes?

Authors’ response: Thank you for your comment, the sentence was modified as suggested.

“To examine the perceived impact of feedback on attainment of learning outcomes, including knowledge acquisition, skill development, and attitudes, to understand the educational value of feedback in the context of pharmacy education.” (line 116)

Reviewer 1: -line 95: a case is not made for why MISCA was chosen. Also it should be spelled out and referenced here. This is better explained in 177-184 however no references are provided in terms of its validated use in other feedback studies which would be helpful to the reader.

Authors’ response: The sentences were revised and adjusted according to your suggestion to make clearer to readers.  And this will mark the first review utilizing the MISCA model; no other feedback studies have incorporated this model before.

“Moreover, this review explores feedback based on feedback metatheories and an integrative feedback model, including five components: message, implementation, student, context, and agents (MISCA) [17]. The MISCA model facilitates a comprehensive exploration of all aspects of feedback. Each component plays an important role in understanding the dynamics of feedback processes. Considering the interrelated components, the integrative MISCA model offers a comprehensive approach to exploring the feedback process and its impact on student learning and performance.” (line 75)

METHODS

Reviewer 1: -line 116: what is the PCC model?

Authors’ response: The PCC model, stands for Population/Participant, Concept, and Context, is a framework commonly used in scoping reviews to guide the formulation of research questions and search strategies. It defines the key elements of the review by specifying the population of interest, the concepts or variables being studied, and the context in which the research is conducted. This model provides a structured approach to identifying relevant studies and ensures that the scope of the review is well-defined.

The sentence was revised and adjusted to make clearer to readers. 

“The PCC model, stands for Population/Participant, Concept, and Context, is a framework commonly used in scoping reviews to guide the formulation of research questions and search strategies [16].” (line 140)

Reviewer 1: -I am concerned the search strategy detailed in appendix A did not detect more literature on Workplace Based Assessments (WBA) and Competency Based Education (CBE) which are becoming more prominent in pharmacy education, and have a significant focus on feedback in experiential settings

Authors’ response: Yes, we appreciate reviewer’s comment and utilization of these methods of assessment in experiential learning. However, our objective was not to look at different methods of assessments in experiential learning but modes and models of feedback. We hope that the keyword feedback would have identified all relevant articles for the purpose of the scoping review.

Reviewer 1: -line 120: was the focus solely on “perspectives of pharmacy students and residents regarding feedback …” Did you also look at the nature and models of feedback, etc (relating to the key objectives lines 86-96)?

Authors’ response: Yes this is true we looked at other objectives too but in line 120 the focus is related to the participants and participants of this review were pharmacy student and residents.

Reviewer 1: -line 141 excludes peer feedback but this is specifically included in line 86? This also seems to be the focus of (for example) reference 25

Authors’ response: Thank you for your observation. The focus of the included article reference 25 is not solely on peer feedback. The review incorporates feedback from both peers and preceptors in experiential learning settings. The articles that were limited on only peer feedback were excluded.

“studies that focused on feedback from other healthcare professions and limited only to peer feedback were excluded.” (line 171)

RESULTS

Reviewer 1: -line 219: why were 795 records excluded?

Authors’ response: According to PRISMA guidelines, providing reasons for exclusion is required during the full-text screening stage. However, at the initial screening stage, where articles are screened based on titles and abstracts, reasons for exclusion are not typically provided.

Reviewer 1: -lines 249-259: referencing is absent/inconsistent

Authors’ response: Thank you for your comment, the sentence was modified as suggested.

“Only one of the 13 studies provided a clear definition of feedback [24], whereas none of the studies discussed the model of feedback in their manuscript. The theoretical framework used was only mentioned in two of the 13 studies [24, 27].” (lines 282-284)

Reviewer 1: -lines 253 and 255 refer to “this study” and “the other study” but the actual studies being referred to are not clear to the reader

Authors’ response: Thank you for your comment, the sentence was modified as suggested.

“The study by Wilbur (2019) employed the cultural dimension models of Hofstede and Hall to understand the feedback encounters and behaviors described by the students. The other study by Jacob Part 1 (2020) adopted the grounded theory method.” (lines 284-287)

Reviewer 1: -line 258 refers to “perceived impact” which I think equates in the table to “student learning outcomes” but this is unclear

Authors’ response: This has now been fixed.

Reviewer 1: -lines 263-274: the wording in this paragraph is very unclear to the reader. Are these barriers and facilitators to receiving feedback, incorporating feedback, preferences, etc for example? Do they relate to giving and/or receiving feedback? How is “lack of feedback” a barrier to receiving feedback?

Authors’ response: We have reworded the term facilitators and barriers throughout the manuscript. We have now replaced facilitators with enablers of effective feedback delivery and barriers to impediments to feedback efficacy so that it is clearer throughout the manuscript and conveys our message more comprehensively.

Reviewer 1: -table 1 appears to include some studies about feedback from learners relating to their experiential placement rather than a focus on receiving feedback from a preceptor on placement. How does data such as this contribute to an understanding of effective models of feedback?

Authors’ response: There is only one study which used peer feedback and preceptor feedback. To ensure comprehensiveness of reporting we included this study as well.

Reviewer 1: Table 1 described the characteristics of the included studies, these limited discussions on the models of feedback and their comparative effectiveness.

Authors’ response: Unfortunately, we failed to identify studies which have compared the effectiveness of two feedback models. We have suggested this as a future research area in our discussion Section 

Reviewer 1: -table 3: my understanding is the agent (person providing the feedback) in this instance should be the preceptor. In a number of cases the agent is the learner themselves which I think means they provided feedback on the program, rather than feedback to the learner which is the intended focus of this review?

Authors’ response: Thank you for your comment. You're correct I modified the agent in table 3.

DISCUSSION

Reviewer 1: -line 349: the prior studies referred to may need to be referenced?

Authors’ response: The sentence was revised and adjusted to make it clearer to readers. 

Reviewer 1:  -line 355: I do not believe a strong case has been made to support the need for a specific definition of feedback in the pharmacy experiential learning setting. Why would this need to be different to a general definition of feedback, and those used in education of other healthcare professionals?

Authors’ response: Thank you for your comment, the sentence was modified as suggested.

“Feedback in healthcare education differs from general feedback due to the critical nature of the medical environment, complexity of medical knowledge, and professional competence involved. The aim is not to create a new definition but rather to establish a consensus on its application within the healthcare education context. This consensus tailoring feedback to specific skills, interdisciplinary collaboration, and patient-centered care. By fostering agreement on these unique aspects, healthcare organizations can promote continuous learning among the learners and professionals and improve patient outcomes [8].” (Line 416)

Reviewer 1:  -line 359: I do not believe a strong case has been made to support the assertion that studies need to incorporate the MISCA model

Authors’ response: Thank you for your comment, the sentence was modified as suggested.

“The MISCA model is comprehensive framework for understanding feedback dynamics and facilitates a thorough investigation of factors influencing feedback effectiveness, including message content, delivery methods, student characteristics, contextual factors, and agent attributes. By identifying barriers and facilitators within feedback processes, the feedback delivery and utilization can be optimized to advance educational practice and enhance student learning outcomes. Moreover, the MISCA model can serve as a framework for future research and practical ideas, guiding the development of evidence-based strategies to improve feedback practices in educational settings [17].” (line 426)

Reviewer 1: -line 367: what does “this model” refer to? MISCA?

Authors’ response: The sentences were revised and adjusted according to your suggestion earlier. 

“Moreover, this review explores feedback based on feedback metatheories and an integrative feedback model, including five components: message, implementation, student, context, and agents (MISCA) [17]. (line 75)

Reviewer 1:  -line 369: I do not disagree with this statement but it is somewhat out of place in this paper. One example is given, a better inclusion might be a discussion of what models are available and suitable to use in this situation, with references in support?

Authors’ response: We have now reworded the whole paragraph

Reviewer 2 Report

Comments and Suggestions for Authors

This scoping review on feedback in the experiential setting in pharmacy education is of importance. My suggestions for improvement follow:

1. Title: consider retitle to better reflect the content of the paper, such as: Providing Feedback in Experiential Learning in Pharmacy Education: A Scoping Review.  (Remove Residents from title, because the focus of the paper is pharmacy students... it is ok to still include what you have on residents, I just wouldn't showcase the word in the title). 

2. Key words: perhaps include pharmacy residents, since you are mentioning them at times in the paper.

3. Abstract: line 12: "challenges faced by pharmacy students during experiential learning";  this is not clear, as I originally interpreted this as problems students encounter, such as difficulty with transportation to the site, parking at the site, challenging personalities at the site, etc., until I read the paper and learned otherwise. Indeed, the intent of the authors is that this word signifies a perceived lack of utility or benefit of feedback received. Please reword (i.e., do not use the words "challenges faced") to clarify this term, in the abstract and throughout the manuscript.

4. Abstract: line 22: "Facilitators" and "Barriers".  These terms are misleading and not intuitive.  Upon reading the paper, I believe the authors intent for facilitators is "beneficial feedback", and barriers is the opposite, or "limited benefit of the feedback".  Please think about the terms that you will use to best describe the feedback: look at your reference 13 by Bing-You for inspiration. Then please reword (i.e., replace the terms facilitators and barriers) in the Abstract and throughout the manuscript (including the Table titles). 

5. Abstract Results: lines 22 and 23.  You have far more interesting and important results than stated in these 2 lines (refer to 3.2, lines 264 - 274).  While there are word count limits for the Abstract, try to reword the Abstract to allot more words to the Results, to showcase more than what you have currently provided.  While the Abstract is just that, an abstraction of the paper, it does set the tone and draws the reader to (or away) from the paper, and needs to be well-written.  You have many sections of the paper that are written better than the abstract, and you can abstract from those. 

6. Abstract Conclusions: line 24 "Our findings highlight the importance of incorporating structured feedback during EL." I do not believe your findings support this statement.  You state the conclusions of your study quite well in the text (see lines 403-405).  I suggest you reword using some of those words, or something like: ...highlights the importance of preceptor education to facilitate effective, high-quality feedback to pharmacy learners.

7. Results line 242: 6/13 (46%) is not a majority (a majority would be >50%).  Perhaps reword to something like: "the country which produced the most studies was the U.S. (6/13, 46%).

8. Appendix A: suggest placing in Supplemental materials.

Comments on the Quality of English Language

The grammar is correct, but the word choices are not always on target, making it difficult to follow -especially in regards to the abstract, which orients the reader to the paper.  For example, in the abstract,:  "...challenges faced by pharmacy students during experiential learning...."  I would interpret that as problems students encounter, such as difficulty with transportation to the site, parking at the site, challenging personalities at the site, etc.  Yet the intent of the authors' use of this word is perceived lack of utility or benefit of feedback received.  Another term used in the abstract and throughout is "facilitator", which would mean someone who enables people to work together, etc., when the authors intended that word to mean beneficial feedback.

Author Response

Reviewer’s comments 2:

This scoping review on feedback in the experiential setting in pharmacy education is of importance. My suggestions for improvement follow:

Reviewer 2: 1. Title: consider retitle to better reflect the content of the paper, such as: Providing Feedback in Experiential Learning in Pharmacy Education: A Scoping Review.  (Remove Residents from title, because the focus of the paper is pharmacy students... it is ok to still include what you have on residents, I just wouldn't showcase the word in the title). 

Authors’ response: We have now changed the title.

Reviewer 2: 2. Key words: perhaps include pharmacy residents, since you are mentioning them at times in the paper.

Authors’ response: Added to the keywords as suggested.

“Keywords: feedback, pharmacy education, pharmacy student, pharmacy resident, preceptors, experiential learning.” (line 26)

Reviewer 2: 3. Abstract: line 12: "challenges faced by pharmacy students during experiential learning";  this is not clear, as I originally interpreted this as problems students encounter, such as difficulty with transportation to the site, parking at the site, challenging personalities at the site, etc., until I read the paper and learned otherwise. Indeed, the intent of the authors is that this word signifies a perceived lack of utility or benefit of feedback received. Please reword (i.e., do not use the words "challenges faced") to clarify this term, in the abstract and throughout the manuscript.

Authors’ response: We have now reworded the statement and taken the word challenges out.

Reviewer 2:  4. Abstract: line 22: "Facilitators" and "Barriers".  These terms are misleading and not intuitive.  Upon reading the paper, I believe the authors intent for facilitators is "beneficial feedback", and barriers is the opposite, or "limited benefit of the feedback".  Please think about the terms that you will use to best describe the feedback: look at your reference 13 by Bing-You for inspiration. Then please reword (i.e., replace the terms facilitators and barriers) in the Abstract and throughout the manuscript (including the Table titles). 

Authors’ response: We have now reworded both of the terminologies so that they are fit for purpose. We have now replaced facilitators with enablers of effective feedback delivery and barriers to impediments to feedback efficacy so that it is clearer throughout the manuscript and conveys our message more comprehensively.

Reviewer 2:  5. Abstract Results: lines 22 and 23.  You have far more interesting and important results than stated in these 2 lines (refer to 3.2, lines 264 - 274).  While there are word count limits for the Abstract, try to reword the Abstract to allot more words to the Results, to showcase more than what you have currently provided.  While the Abstract is just that, an abstraction of the paper, it does set the tone and draws the reader to (or away) from the paper, and needs to be well-written.  You have many sections of the paper that are written better than the abstract, and you can abstract from those. 

Authors’ response: Thank you for your comment, the sentence was modified as suggested.

“Facilitators of feedback included timely feedback (n=6, 46%), feedback provided in a goal-oriented and objective manner (n=5, 40%), and student-specific feedback (n=4, 30%). Barriers included providing extremely positive feedback and lack of constructive criticism.” (Line 22)

Reviewer 2:  6. Abstract Conclusions: line 24 "Our findings highlight the importance of incorporating structured feedback during EL." I do not believe your findings support this statement.  You state the conclusions of your study quite well in the text (see lines 403-405).  I suggest you reword using some of those words, or something like: ...highlights the importance of preceptor education to facilitate effective, high-quality feedback to pharmacy learners.

Authors’ response: Thank you for your comment, the sentence was modified as suggested.

“Conclusions: Our findings highlight the importance of feedback model implementation in pharmacy education and preceptor training programs to ensure effective and quality feedback to pharmacy students.” (Line 25)

Reviewer 2:  7. Results line 242: 6/13 (46%) is not a majority (a majority would be >50%).  Perhaps reword to something like: "the country which produced the most studies was the U.S. (6/13, 46%).

Authors’ response: Thank you for your comment, the sentence was modified as suggested.

“Almost half of the included studies were conducted in the USA (n=6, 46%)” (lines 20,272)

Reviewer 8. Appendix A: suggest placing in Supplemental materials.

Authors’ response: We will explore if this can be added as supplemental material.

Comments on the Quality of English Language

Reviewer’s comment: The grammar is correct, but the word choices are not always on target, making it difficult to follow -especially in regards to the abstract, which orients the reader to the paper.  For example, in the abstract,:  "...challenges faced by pharmacy students during experiential learning...."  I would interpret that as problems students encounter, such as difficulty with transportation to the site, parking at the site, challenging personalities at the site, etc.  Yet the intent of the authors' use of this word is perceived lack of utility or benefit of feedback received.  Another term used in the abstract and throughout is "facilitator", which would mean someone who enables people to work together, etc., when the authors intended that word to mean beneficial feedback.

Authors’ response: We have now replaced the terms facilitators and barriers with enablers of effective feedback delivery and impediments to feedback efficacy respectively.

Round 2

Reviewer 2 Report

Comments and Suggestions for Authors

Nice job with revising this paper.  It reads much better now.